# DEEP CONVOLUTION FOR IRREGULARLY SAMPLED TEMPORAL POINT CLOUDS

## ABSTRACT

We consider the problem of modeling the dynamics of continuous spatial-temporal processes represented by irregular samples through both space and time. Such processes occur in sensor networks, citizen science, multi-robot systems, and many others. We propose a new deep model that is able to directly learn and predict over this irregularly sampled data, without voxelization, by leveraging a recent convolutional architecture for static point clouds. The model also easily incorporates the notion of multiple entities in the process. In particular, the model can flexibly answer prediction queries about arbitrary space-time points for different entities regardless of the distribution of the training or test-time data. We present experiments on real-world weather station data and battles between large armies in StarCraft II. The results demonstrate the model's flexibility in answering a variety of query types and demonstrate improved performance and efficiency compared to state-of-the-art baselines.

## 1 INTRODUCTION

Many real-world problems feature observations that are sparse and irregularly sampled in both space and time. Weather stations scattered across the landscape reporting at variable rates without synchronization; citizen-science applications producing observations at the whim of individuals; or even opportunistic reports of unit positions in search-and-rescue or military operations. These sparse and irregular observations naturally map to a set of discrete space-time points – forming a spatio-temporal point cloud representing the underlying process. Critically, the dynamics of these points are often highly related to the other points in their spatio-temporal neighborhood.

Modelling spatio-temporal point clouds is difficult with standard deep networks which assume observations are dense and regular – at every grid location in CNNs, every time step in RNNs, or both for spatio-temporal models like Convolutional LSTMs (Xingjian et al., 2015). While there has been work examining irregularly sampled data through time (Rubanova et al., 2019; Shukla & Marlin, 2018) and in space (Wu et al., 2019), modeling both simultaneously has received little attention (Choy et al., 2019). This is due in part to the difficulty of scaling prior solutions across both space and time. For instance, voxelization followed by sparse convolution (Choy et al., 2019) or dense imputation (Shukla & Marlin, 2018) now face a multiplicative increase in the number of cells.

Rather than forcing irregular data into dense representations, an emerging line of research treats spatial point-clouds as first-class citizens (Qi et al., 2017a;b; Su et al., 2018; Xu et al., 2018). Several works directly extend 2D convolutions to point clouds (Simonovsky & Komodakis, 2017; Wang et al., 2019; Hermosilla et al., 2018), with (Wu et al., 2019) being the first that allows efficient exact computation of convolution with dozens of layers. In this work, we build on this line of research to model spatio-temporal point clouds. Specifically, we extend the work of Wu et al. (2019) with an additional module to reason about point representations through time.

Our new model, TemporalPointConv (TPC), is a simple but powerful extension that can learn from an arbitrary number of space-time points. Each layer in TemporalPointConv updates the representation of each point by applying two operators in sequence – one that considers the spatial neighborhood in a narrow temporal window and another that models how this spatial representation changes over time. By factorizing the representation update into separate spatial and temporal operators, we gain significant modeling flexibility. Further, by operating directly on point clouds, we can predict observations at arbitrary space-time, regardless of the distribution of observations.

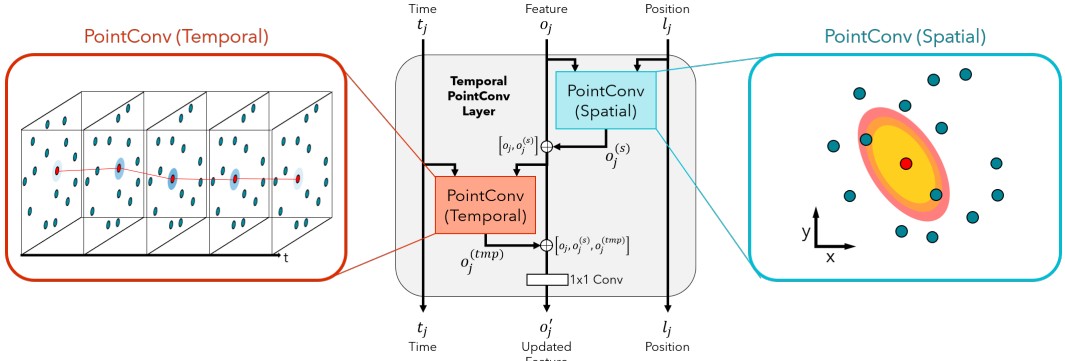

Figure 1: **TemporalPointConv** operates on unsynchronized sets of spatio-temporal samples by applying two point-based convolutional operators in sequence, each of which exploits separate notions of either spatial or temporal locality.

We demonstrate TemporalPointConv on two distinct problems: 1) predicting future states of a custom Starcraft II environment involving battles between variable-sized groups, and 2) predicting the weather at stations distributed throughout the state of Oklahoma. Further, we show the utility of these networks in identifying damaged or anomalous weather sensors after being trained exclusively on the associated prediction problem. The results show that TemporalPointConv outperforms both state of the art set functions and a discrete sparse convolution algorithm in terms of raw performance, ability to detect anomalies, and generalization to previously unseen input and query distributions.

## 2 RELATED WORK

Xingjian et al. (2015) gives an early approach to spatio-temporal modeling via convolution by incorporating a standard convolutional structure into the latent memory of an LSTM. This approach is appropriate for situations where the data is regularly sampled in both space and time, which is different from our setting. Interaction networks (Battaglia et al., 2016) and related approaches allow for modeling sets of interacting objects or points over time, with an original motivation to model physics processes. These models are more flexible in their modeling of spatial relationships among points. However, there is an assumption of uniform temporal sampling, which is violated in our setting.

A significant amount of work on spatio-temporal modeling for non-uniform spatial sampling uses Graph Convolutional Networks (GCNs) for modeling spatial interactions. For example, Li et al. (2018b) used a GCN followed by an RNN and Yu et al. (2018) used GCNs for spatial correlation and temporal convolution for temporal correlations. They require sampling at continuous temporal intervals and did not deal with generalization outside the fixed given graph. Rather, our approach generalizes to any spatio-temporal point outside of the training data. Yao et al. (2019) introduces an attention model to deal with dynamic spatial relationships, however this is only possible for the dense CNN version in their paper, whereas their version with irregular spatial sampling utilizes the GCN and shares the same issues with the above GCN approaches.

PointNet (Qi et al., 2017a) sparked significant interest in networks for 3D Point cloud processing. A number of networks have been proposed (Qi et al., 2017a;b; Su et al., 2018; Xu et al., 2018) with the highest performing using either sparse convolutional networks (Graham & van der Maaten, 2018; Choy et al., 2019) or point convolutional networks (Wu et al., 2019; Thomas et al., 2019). Set networks, such as DeepSets (Zaheer et al., 2017b), are similar to PointNet (Qi et al., 2017a) with neither explicitly considering neighborhood information of elements/points, making them less powerful than convolutional methods. Recently, Horn et al. (2020) proposed a set network approach for non-uniform time-series prediction, which encodes time into the feature vector of points. Our experiments show that this approach is outperformed by our convolutional method.

Sparse convolutional networks are similar to dense volumetric convolutional networks that use a regular grid to discretize space-time, but they are only computed at locations with occupied points. Minkowski networks (Choy et al., 2019) is a sparse convolutional network that models spatio-

temporal correlations by concatentating the spatial location and time for each point sample into a 4D tesseract. It is thus sensitive to an appropriate resolution for the discretization since excess sparsity can result in empty neighborhoods and trivial convolutions, and too coarse a resolution may result in an inaccurate representation of the data. Furthermore, the approach has difficulties accounting for the case where points should be treated as moving entities themselves.

On the other hand, point convolutions discretize 3D volumetric convolutions on each point directly and hence easily generalize to the entire space under irregular sampling density. Early versions (Simonovsky & Komodakis, 2017; Hermosilla et al., 2018; Wang et al., 2019) require explicit discretization hence cannot scale to large networks. Recently, PointConv (Wu et al., 2019) proposes an equivalent form that avoids explicit discretization and significantly improved scalability. However, so far it has been applied only to static point clouds. Our work builds on PointConv, by extending it in the temporal direction and demonstrating that space-time convolutions can be effectively learned and used for modeling and anomaly detection.

On the temporal side, a significant amount of recent state-of-the-art were based on point processes which studies time series models from a statistical perspective (Du et al., 2016; Li et al., 2018a; Zuo et al., 2020; Zhang et al., 2019). These support irregular temporal sampling, but generally do not consider the spatial correlation among points.

## 3 PROBLEM SETUP

We consider extrapolative tasks in which the value at new locations must be inferred from existing observations. Let $P$ be a spatio-temporal point cloud with each individual point $p_j \in P$ defined as $p_j = (l_j, t_j, o_j)$ where $p_j$ exists at location $l_j$ at time $t_j$ and has associated features $o_j$ (e.g. temperature and humidity values for a weather station). Further, let $Q$ be a set of query locations at which the model is to make predictions given $P$. For example, a forecasting model might be given queries $q_k = (l_k, t_k)$ for locations in the future and be tasked with predicting the corresponding features $o_k$ representing the desired properties to be predicted. We place no restrictions on the regularity of either $P$ or $Q$ such that this corresponds to a setting where both input and output may be sparse and irregularly sampled through space and time. Further, query points may be in the future, the past, or concurrent with those in $P$ – corresponding to weather forecasting, backcasting, or nowcasting respectively. We aim to train models that can accurately answer queries as represented via training set of point-cloud / query-set pairs $D = \{(P_i, Q_i)\}_{i=1}^N$.

## 4 TEMPORAL POINTCONV ARCHITECTURE

Given a spatio-temporal point-cloud containing points $p_j = (l_j, t_j, o_j)$, a Temporal PointConv layer is an operator that produces an updated point representation $p_j' = (l_j, t_j, o_j')$ for each point. The updated feature representation $o_j'$ incorporates information from a spatio-temporal neighborhood around $p_j$. This is accomplished by applying two point-based convolutional operators in sequence for each point – first a spatial PointConv over points within a narrow temporal band, and then a temporal PointConv over points within a narrow spatial band. These Temporal PointConv layers can be stacked to arbitrary depth. Below we give background on PointConv and describe our model.

### 4.1 PRELIMINARIES: POINTCONV

PointConv is based on the idea of discretizing continuous convolution on irregularly sampled points:

$$Conv(P, \mathbf{p}_0; \mathbf{w}, d(\cdot, \cdot)) = \sum_{\mathbf{p}_i \in \mathcal{N}_d(\mathbf{p}_0)} \langle \mathbf{w}(\mathbf{p}_i - \mathbf{p}_0), \mathbf{o}_i \rangle \tag{1}$$

where $P$ is a point cloud with features at each point, $\mathbf{w}(\cdot)$ is a vector-valued weight function of the positional difference between a point $\mathbf{p}_i$ in the neighborhood $\mathcal{N}_d$ of a centroid $\mathbf{p}_0$, defined by a metric $d$, and $\mathbf{o}_i$ is the input features at $\mathbf{p}_i$. $\mathbf{w}(\cdot)$ can be learned with a neural network (Simonovsky & Komodakis, 2017). PointConv (Wu et al., 2019) introduces an equivalent form so that $\mathbf{w}$ does not need to be computed explicitly, saving computation and memory.

This approach is flexible since $\mathbf{w}(\cdot)$ as a function can apply to any point in the space of $P$, hence convolution can be computed over any irregularly sampled neighborhood $\mathcal{N}_d$. We note that this even holds when we did not have any feature at $\mathbf{p}_0$, since a neighborhood can still be found even in this

case and eq. (1) can still be used. Previously, PointConv has only been used in spatial domains in cases where $\mathbf{p}_0$ has features associated with it. In this paper we generalize it to spatio-temporal neighborhoods and to $\mathbf{p}_0$ that are featureless query points.

For expositional clarity, we denote PointConv as an operator that transforms a feature-augmented point-cloud $P$ into a new point-cloud $P'$ consisting of points at target locations $Q$ with eq. (1): $P' = PointConv(P, Q; d(\cdot, \cdot))$, where we will omit $Q$ if $Q = P$.

## 4.2 TEMPORAL POINTCONV

Given a spatio-temporal point-cloud $P_{in} = \{(l_j, t_j, o_j^{(in)})|j\}$ and set of queries $Q$, the Temporal PointConv operations considers the relative position from each query to the elements of $P_{in}$ and their representative features to produce a set of predictions $X$ corresponding to the query set $Q$.

**Spatial Convolution.** First, each point's feature is updated based on the spatial neighborhood of temporally co-occurring points. However, as the points may be irregularly spaced in time, there may be no points that precisely co-occur. We instead consider those in a fixed window of time. Thanks to the flexibility of PointConv operations, we describe this by defining the piece-wise distance function:

$$d_{spatial}(p_i, p_j) = \begin{cases} || l_i - l_j ||_2 & \text{if } |t_i - t_j| \leq \epsilon_t \\ \infty & \text{otherwise} \end{cases}. \tag{2}$$

We then apply a PointConv operator to update features: $P_{spatial} = PointConv(P_{in}; d_{spatial})$, where each point in $P_{spatial}$ has updated feature $(l_i, t_i, o_i^{(s)})$.

**Temporal Convolution.** We then perform an analogous operation through time. We would like to consider the past and future of each point; however, this requires determining correspondence between points through time. If the underlying point-cloud represents static points such as weather stations, this can simply be based on a small spatial window. If the points correspond to known entities that are moving, we instead assume tracking and can use those entity labels to determine temporal neighborhoods each consisting exclusively of a single entity's samples throughout time. For clarity, we present the distance function for the first case below:

$$d_{temporal}(p_i, p_j) = \begin{cases} || t_i - t_j ||_2 & \text{if } || l_i - l_j ||_2 \leq \epsilon_s \\ \infty & \text{otherwise} \end{cases}. \tag{3}$$

Before applying the temporal PointConv, we first apply a residual connection for each point, concatenating the input and spatial features. We denote this as $P_{res} = \{(l_j, t_j, [o_j^{(in)}, o_j^{(s)}]) \mid j\}$ where $[\cdot, \cdot]$ denotes concatenation. As before, we apply a PointConv operator with kernels defined only over differences in time as: $P_{temporal} = PointConv(P_{res}; d_{temporal}(\cdot, \cdot))$, where $P_{temporal} = \{(l_j, t_j, o_j^{(tmp)}])|j\}$.

**Combined Representation.** To compute the final output point-cloud, we concatenate the original, spatial, and temporal representations and transform them through an MLP $f$ such that

$$P_{out} = \{(l_j, t_j, f([o_j^{(in)}, o_j^{(s)}, o_j^{(tmp)}]) \mid j\}. \tag{4}$$

We denote multiple stacked layers via $P^{(d+1)} = TemporalPointConv(P^{(d)})$.

## 4.3 EXTRAPOLATING TO NEW POINTS

After applying one or more layers of Temporal PointConv as described above, we apply one final *query PointConv* to the latent spatio-temporal point cloud $P_{out}$ resulting from this encoding process. For this, we define a new problem-dependent query distance function $d_{query}(\cdot, \cdot)$, which could be $d_{spatial}$, $d_{temporal}$, or a combination of both. This enables us to calculate a corresponding latent feature $y$ for the each query point.

$$Y = PointConv(P_{out}, Q; d_{query}(\cdot, \cdot)) \tag{5}$$

Finally, we apply an MLP $g$ to transform each latent query representation into a final predictions $X = \{g(o_y)|y \in Y\}$ corresponding to the set of queries $Q$.

## 5 EXPERIMENTS

We consider two problem domains for our experiments which we describe below.

**Starcraft II.** To evaluate TemporalPointConv on entity-based dynamics, we designed a custom Starcraft II scenario in which two opposing armies consisting of random numbers of three distinct unit types are created and then fight on a featureless battlefield. Each episode is allowed to run without any external influence until one team has been eliminated or the time limit expires. This allows us to learn the dynamics of a battle between a large group of units without any confounding factors such as player inputs. We use the PySC2 library (Vinyals et al., 2017) to record regular observations of the game state as each episode plays out.

We use these regularly sampled episode histories to generate individual training examples. Specifically, we select a 'reference timestep' $t$ within the episode, sample a set of 'history offsets' $H$ from a provided history distribution, and a set of 'query offsets' $R$ from a provided query distribution. We collect unit properties corresponding to these sampled relative time steps to serve as point features. We determine the prediction targets with the same procedure using the sampled query offsets. This procedure is used to sample an arbitrary number of training examples from the set of episode histories by varying the reference timestep $t$ and re-sampling the history and query offsets as desired. Following this procedure on our dataset of 92,802 episodes yields 2.5 million training examples.

We define the 'property loss' for a unit state prediction as the sum of the mean squared error of each of the unit's predicted numeric properties (i.e. health, shields, position) and the cross entropy loss of the unit's predicted categorical properties (i.e. orientation). Similarly, the 'alive loss' is the cross entropy loss between the network's alive/dead prediction values and a flag indicating if the unit was present and alive in the given timestep. We then define the total loss for a set of unit state predictions as the sum of the alive loss for all units and with property loss for every unit that is actually alive at the given timesteps. This additional condition is necessary due to the fact that dead units do not have recorded properties we can use to determine property loss.

As PySC2 assigns a unique, consistent ID to each unit which provides perfect tracking across all timesteps, we use an entity-based temporal distance function when instantiating the query PointConv layer for this problem as described in section 4.2 above.

**Weather Nowcasting.** To evaluate the ability of the TemporalPointConv architecture to reason about spatio-temporal dynamics, we derive weather nowcasting problems from a dataset of weather conditions as recorded by weather stations throughout Oklahoma. The original dataset consists of weather sensor readings from each weather station every five minutes throughout the entirety of the year 2008, associated quality metrics for each sensor in each reading, and metadata about each weather station such as its position and local soil properties.

10% of the weather stations are randomly selected to be held out as test stations and excluded from the training process, while the remaining 90% are used to generate problems for training. We derive training problems from the larger dataset by selecting a time point $t$ and randomly selecting 10% of the remaining training stations to be targets. All non-target training station readings and their associated station metadata within the hour preceding $t$ are collected as input weather data. Any sample within the collected data with an associated quality metric indicating a malfunctioning or missing sensor is discarded. Furthermore, we randomly discard an additional 20% of the remaining samples to decrease the level of time synchronization in the input. Following this procedure on our dataset of weather sensor readings results in over 14,000 training examples.

The model is then tasked with predicting weather properties at time $t$ for each of the target stations using the provided input data from the preceding hour. Specifically, the networks are evaluated on their ability to predict the relative humidity, air temperature, air pressure, and wind speed at each specified target location. We define the prediction loss as the sum of the mean square error between the network's prediction for each of these properties and the actual recorded values. Due to the large difference in magnitudes between these readings, normalize each prediction and target measurement value such that the 10th percentile value to 90th percentile value of that measurement within the entire dataset is mapped to the range [0, 1]. This prevents the training process from naturally favoring measurements with a much higher average magnitude than the others.

As our queries for this problem are purely spatial, we use the spatial distance function eq.(2) as the query distance function when instantiating the query PointConv layer for this problem.

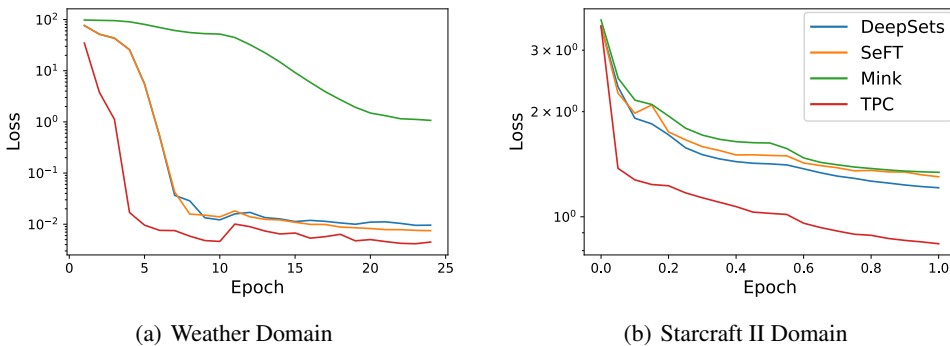

(a) Weather Domain                    (b) Starcraft II Domain

Figure 2: Validation performance throughout the training process averaged across four runs.

## 5.1 BASELINE IMPLEMENTATIONS

**Set Functions for Time Series & DeepSets.** Our Temporal PointConv architecture leverages Point-Conv as a convolution-equivalent set function. We can evaluate this choice by replacing each Point-Conv module with a different set function, such as DeepSets (Zaheer et al., 2017a) or Set Functions for Time Series (SeFT) (Horn et al., 2020). Whereas PointConv takes as input a set of point locations and a set of point features, SeFT and DeepSets only consume a single set of features. However, the neighborhood and distance function mechanisms introduced for Temporal PointConv can still be applied. Therefore, we evaluate the other set functions by simply replacing each instance of $PointConv(P)$ with $SeFT(\{[l_i, t_i, o_i]||i\})$ or $DeepSets(\{[l_i, t_i, o_i]||i\})$.

**Minkowski Networks.** We evaluate Minkowski networks (Choy et al., 2019) by replacing each spatial-temporal PointConv step with a Minkowski convolution layer that operates on the combined spatio-temporal vector space inhabited by the raw input samples. This necessarily requires discretizing said vector space into a sparse voxel grid. We choose a voxel resolution of 6km for the weather domain, and 0.05 in game units for the starcraft domain. We use nVidia's MinkowskiEngine codebase to provide the Minkowski convolution implementation.

We trained Temporal PointConv (TPC), Set Function for Time Series (SeFT), DeepSets, and Minkowski networks instantiated with the hyperparameter settings described in appendix B on both the Starcraft II and weather nowcasting domains. For the Starcraft II domain, models were trained for one epoch (owing to the massive size of the generated Starcraft II dataset), whereas for weather nowcasting they were trained for 24 epochs. All networks were trained with a cosine learning rate decay with warm restarts configured such that the learning rate cycles from its maximum value to its minimum three times throughout each training run.

## 5.2 RESULTS

**Dynamics Prediction Accuracy.** To evaluate prediction accuracy, three of each model were trained on both domains. Unless otherwise specified, the Starcraft history distribution was set to be a uniform distribution over $[-10, -1]$ and the query distribution was set to fixed time offsets $\{1, 2, 4, 7\}$. Figure 2 shows the validation loss for each model throughout training, and tables 1 and 2 show in detail the average error across each individual query the final trained networks predict for the test datasets. Our results show that TPC is significantly more accurate than the baseline algorithms, es-

Table 1: **Weather Nowcasting.** Mean per-query prediction loss and 95% confidence interval for each target attribute across the test dataset. Normalized loss is calculated as described in section 5.

| Model | Rel. Humidity | Air Temp. | Wind Speed | Air Pressure | Normalized Loss |
|-------|---------------|-----------|------------|--------------|-----------------|
| SeFT | 22.97 $_{\pm57.36}$ | 1.65 $_{\pm3.46}$ | 0.87 $_{\pm1.85}$ | 5.40 $_{\pm11.16}$ | 0.0299 $_{\pm0.0003}$ |
| DeepSets | 25.07 $_{\pm57.78}$ | 2.36 $_{\pm4.11}$ | 0.99 $_{\pm2.15}$ | 15.88 $_{\pm40.69}$ | 0.0383 $_{\pm0.0004}$ |
| Minkowski | 260.72 $_{\pm401.69}$ | 62.20 $_{\pm94.15}$ | 7.75 $_{\pm12.32}$ | 9668.62 $_{\pm6670.14}$ | 4.2803 $_{\pm0.0202}$ |
| TPC (Ours) | 9.75 $_{\pm33.67}$ | 1.43 $_{\pm2.41}$ | 0.54 $_{\pm1.26}$ | 3.75 $_{\pm4.89}$ | 0.0179 $_{\pm0.0002}$ |

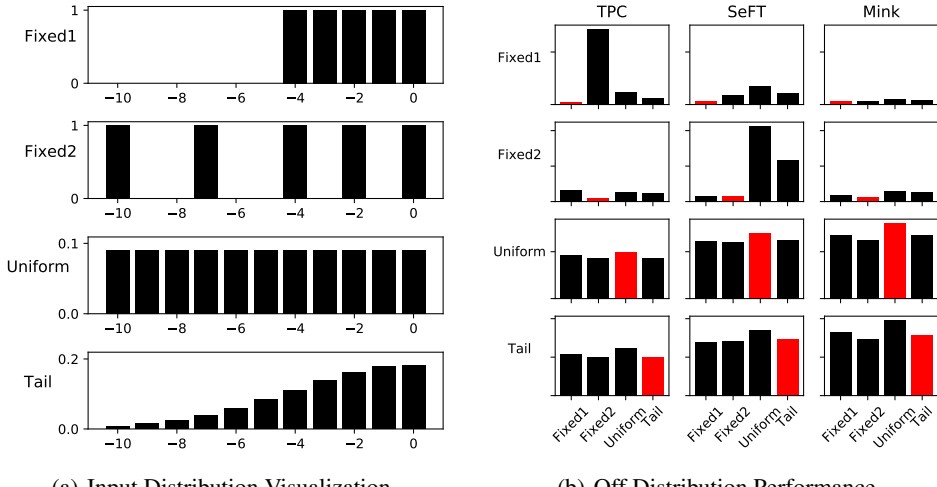

(a) Input Distribution Visualization      (b) Off Distribution Performance

Figure 3: **Input Distribution Comparison.** 3(a) shows the probability of selecting each timestep as input during training for each distribution type. 3(b) depicts the relative performance of networks trained on each input distribution when evaluated across all four. Note that Y-axes are scaled by row.

pecially on the Starcraft II unit state prediction problem. In all cases, the Minkowski network was unable to outperform either of the set function-based models, and in the weather nowcasting domain it consistently failed to find a good solution, as indicated by the loss orders of magnitude higher than the set function approaches. We believe this failure is due to the difficulty of selecting a suitably sized kernel and voxelization resolution for a spatio-temporal problem at the scale of an entire state. We were unable to increase the size of the kernel without driving the network's parameter count prohibitively high, and we were unable to decrease the resolution of voxelization without starting to 'lose' a significant number of weather stations which would be occupying the same cell. This result suggests that applying 'true' point cloud convolution that directly exploits sample positions is preferable for these domains, as opposed to discretizing or voxelizing the samples' locations so that a traditional fixed-size filter convolution such as Minkowski networks can be applied.

**Impact of Train and Test Distributions.** We investigate the robustness of TPC to a change in the distribution of input samples or query points. Since the TPC architecture is completely decoupled from the distribution of the input samples, we can accomplish this comparison by simply defining several distribution types, training a model with each type of input distribution on the Starcraft II domain, and comparing the results after evaluating each trained model across each of the input distribution types selected for evaluation.

We selected four input distributions for evaluation: Two 'fixed' distributions that always return the same set of time offsets, the uniform distribution over the range $[-10, 0]$, and half of a normal distribution over the range $[-10, 0]$. Figure 6 visualizes the difference between these distributions, and presents a bar chart plotting the average loss when each model is evaluated on each distribution type. In all cases, the query distribution was kept constant and fixed. The results show that TPC and SeFT trained on fixed distributions perform poorly when evaluated on any distribution it was not trained on, while the Minkowski network suffers much less of a penalty despite worse absolute performance. Alternatively, the networks trained on the uniform and normal distributions suffer much less degradation when switching to different input distributions. The only case with a noticeable

Table 2: **Starcraft II.** Mean per-query prediction loss and 95% confidence interval of each algorithm for each target attribute on the test set.

| Model | Position | Health | Shield | Orientation | Alive | Total Error |
|---|---|---|---|---|---|---|
| SeFT | 0.139 ±0.4151 | 0.018 ±0.0464 | 0.005 ±0.0177 | 1.260 ±1.1215 | 0.175 ±0.4472 | 1.600 ±0.0016 |
| DeepSets | 0.127 ±0.4158 | 0.014 ±0.0393 | 0.004 ±0.0156 | 1.218 ±1.1646 | 0.158 ±0.4291 | 1.524 ±0.0017 |
| Minkowski | 0.666 ±0.8572 | 0.131 ±0.1929 | 0.021 ±0.0532 | 1.865 ±0.5880 | 0.464 ±0.5166 | 3.141 ±0.0013 |
| TPC (Ours) | 0.083 ±0.3502 | 0.006 ±0.0187 | 0.002 ±0.0084 | 1.017 ±1.3591 | 0.102 ±0.3569 | 1.213 ±0.0018 |

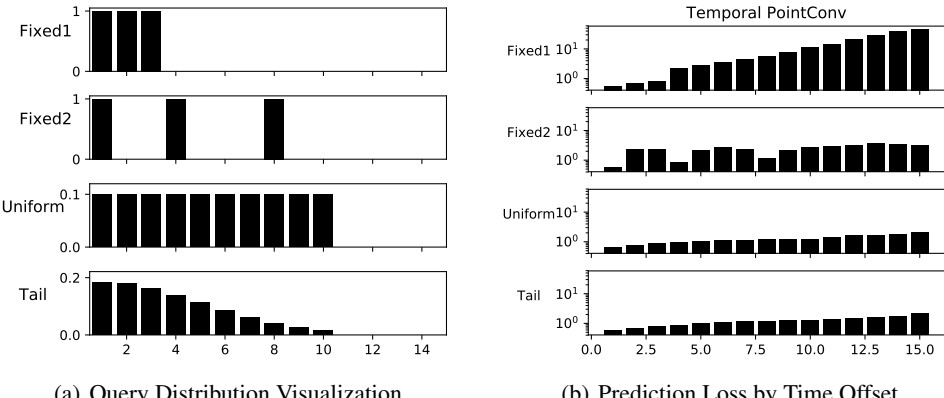

(a) Query Distribution Visualization          (b) Prediction Loss by Time Offset

Figure 4: **Query Distribution Comparison.** 4(a) shows the probability of selecting each timestep as a query during training. 4(b) shows loss on queries by timestep up to $t = 15$ for networks trained on each query distribution.

performance drop is for networks trained on the normal distribution and evaluated on the uniform distribution, which is unsurprising since the normal distribution is biased toward $t = 0$.

We perform a similar experiment to evaluate the behavior of TPC when trained on different query distributions. Figure 4 visualizes the query distributions selected for training alongside a plot of the average loss for each query by their offset from the reference time (e.g. $t = 0$). As before, the models trained on fixed distributions only consistently perform well on the exact query points they were trained on, with the model trained on Fixed1 distribution's prediction error rising sharply as the distance from its small cluster of expected query points increases. In contrast, the model trained on the variable distributions saw a relatively small increase in prediction error, even for query points that are outside of the range of query points it was trained on. This suggests that the ability to train the TemporalPointConv architecture on randomized input and query distributions is key to enabling it to generalize well across timesteps and behave reasonably in off-distribution scenarios.

**Application to Anomaly Detection.** We now consider the utility of our TPC model for anomaly detection, where the goal is to detect which samples in a temporal point cloud are anomalous. We focus on the weather dataset, where anomalies correspond to broken sensors. We introduce anomalies to the set of test station samples by randomly selecting 33% of the stations. For these, we randomly increase or decrease the value of one station property by a factor of 25%. The models are then tasked with predicting each of the test samples' properties given the preceding hour of weather data. Their prediction error on each individual sample is then used as an anomaly score for detection purposes.

As expected based on prior prediction results, TPC significantly outperforms SeFT owing to its superior nowcasting accuracy with an area under receiver-operator curve (AUROC) of 0.927 compared to SeFT's 0.836. The Minkowski network struggles to perform above chance level. See appendix A for the complete ROC curves.

## 6 CONCLUSION

In this work, we proposed a novel extension to the set function PointConv that enables it to be composed with standard deep learning layers to reason about irregularly sampled spatio-temporal processes and calculate predictions for arbitrary domain-specific queries. We show that Temporal-PointConv's ability to directly consume each sample's positional and feature data without down-sampling or discretization enables it to significantly outperform state of the art sparse convolution algorithms across two complex, meaningfully different domains. Similarly, TemporalPointConv's equivalence to standard convolution enables it to more efficiently reason about relative spatial and temporal relationships than other set functions which are not endowed with these useful properties. These promising results and TemporalPointConv's flexible parameterization suggest that it can be effectively applied to a wide range of problems with an irregular structure that prevents most other deep learning approaches from functioning efficiently.

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

## A ANOMALY DETECTION ROC CURVES

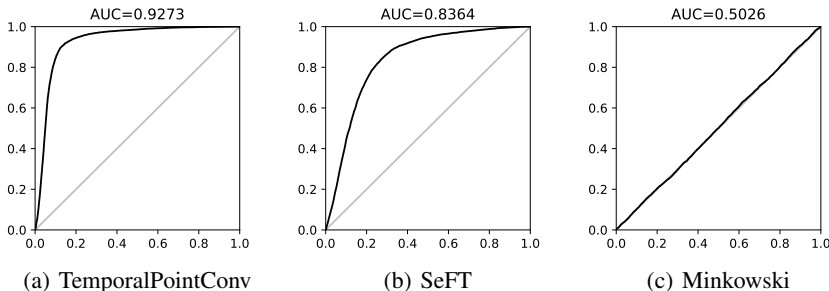

(a) TemporalPointConv     (b) SeFT     (c) Minkowski

Figure 5: ROC curves of each model's prediction error thresholding anomaly detection performance.

## B HYPERPARAMETER SETTINGS

Table 3: Hyperparameter settings used to instantiate each type of model on each type of domain.

| Domain | Starcraft | | | | Weather | | | |
|---|---|---|---|---|---|---|---|---|
| Model | TPC | SeFT | DeepSets | Minkowski | TPC | SeFT | DeepSets | Minkowski |
| PointConv: Weight Network Hidden | 16, 16 | | | | 32, 32 | | | |
| PointConv: C_mid | 32 | | | | 32 | | | |
| PointConv: Final MLP Hidden | 64, 64 | | | | 64, 64 | | | |
| DeepSets: DeepSets Hidden | | 192, 192, 192 | 192, 192, 192 | | | 128, 128, 128 | 192, 192, 192 | |
| DeepSets: Self Attention | | Yes | No | | | Yes | No | |
| Mink: Voxel Resolution | | | | 0.05 | | | | 6000 km |
| Mink: Kernel Size | | | | 21, 21, 17 | | | | 21, 21, 21 |
| Latent Neighborhood Feature Sizes | 16, 32, 32 | 16, 32, 32 | 16, 32, 32 | 16, 32, 32 | 16, 32, 32 | 16, 32, 33 | 16, 32, 33 | 16, 32, 32 |
| TemporalPointConv Encoder Hidden Layers | 32, 64, 64 | 32, 64, 64 | 32, 64, 64 | 32, 64, 64 | 32, 64, 64 | 32, 64, 64 | 32, 64, 64 | 32, 64, 64 |
| Max Neighbors | 8 | 8 | 8 | 8 | 8 | 8 | 8 | 8 |
| Query Latent Size | 64 | 64 | 64 | 64 | 64 | 64 | 64 | 64 |
| Decoder MLP Hidden | 64, 64, 64 | 64, 64, 64 | 64, 64, 64 | 64, 64, 64 | 64, 64, 64 | 64, 64, 64 | 64, 64, 64 | 64, 64, 64 |
| Learning Rate Range | (1e-3, 1e-6) | (3e-4, 1e-5) | (3e-4, 1e-5) | (1e-3, 1e-6) | (1e-3, 1e-7) | (1e-3, 3e-5) | (1e-3, 3e-5) | (1e-3, 1e-6) |
| Optimizer | ADAM | ADAM | ADAM | ADAM | ADAM | ADAM | ADAM | ADAM |
| Parameter Count | 1.3M | 1.3M | 1.0M | 50M | 960k | 745k | 381k | 104M |

## C JOINT SPACE-TIME NEIGHBORHOODS

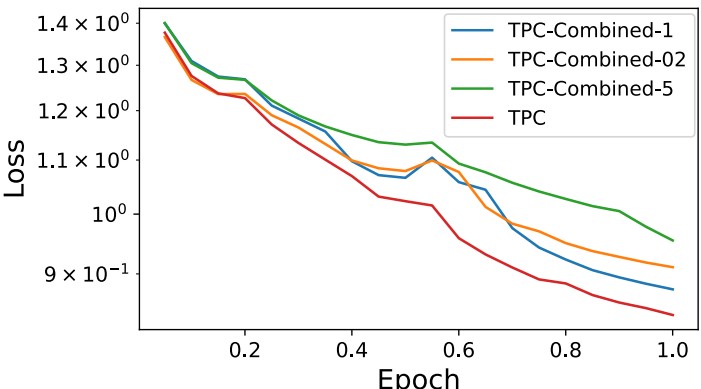

Figure 6: Combined distance function experiment results.

Though TemporalPointConv decomposes spatio-temporal processes into separate 'space' and 'time' neighborhoods, this is not strictly necessary. Space and time could be combined into one single vector space, allowing for a single PointConv layer to jointly consider samples' spatial and temporal distances to determining their local neighborhood.

We investigate this possibility by training TemporalPointConv networks to do exactly that. This requires specifying a space-time distance function which we define as follows: $D_{st} = \sqrt{D_s^2 + xD_t^2}$

where $D_s$ and $D_t$ are spatial and temporal distance functions, respectively. $x$ then represents the tradeoff factor that dictates whether distant spatial samples should be favored over temporally distant samples when constructing a neighborhood.

Specifically, we test three values for $x$ for these 'combined' PointConv models: 0.2. 1, and 5. The results in figure C show that all of the networks with combined spatial-temporal neighborhood functions were outperformed by our approach which considers spatial and temporal relationships separately but sequentially. Additionally, this combined distance function depends on a hyperparamter $x$ which is likely domain-specific and nontrivial to find a good value for. These results validate our decision to treat spatial and temporal distances separately.

