# OpenReview forum: "Deep Convolution for Irregularly Sampled Temporal Point Clouds"
_ICLR.cc/2021/Conference — Reject_

### Official Review · AnonReviewer4 · 2020-10-25
**An extension of PointConv, evaluation is not that convincing.**

**Rating:** 5
**Confidence:** 4

**Review:**

The paper proposes an extension of PointConv for spatial-temporal point cloud modeling. The model can be used for prediction or forecasting and is evaluated on Starcraft II and weather nowcasting.

- The TemporalPointConv follows PointConv and the current work extends this by appending time. I think the paper is heavily based on PointConv which makes the overall novelty limited.

- The temporal convolution either does not track points or relies on entity labels for tracking. As in real and most point-cloud scenarios point labels are not available, the temporal convolution restricts the applications of the proposed method.

- Starcraft II and weather nowcasting are not common applications of point cloud and I thus feel the evaluation is not convincing enough. It would be more appealing to apply the method to other applications such as autonomous driving cars.

- Making weather nowcasting via point clouds has been studied [a]. Discussion and comparison with related works are missing.
[a] CloudLSTM: A Recurrent Neural Model for Spatiotemporal Point-cloud Stream Forecasting

- I think the idea of appending time dimension can also be applied to PointNet++, KPConv, or other operations. Why do we choose PointConv? Is there any problem if the proposed method is based on other operations?

=========Post Rebuttal==========
Because no response is provided, I maintain my original rating.

---

### Official Review · AnonReviewer3 · 2020-10-26
**Potentially interesting results, needs some revision**

**Rating:** 5
**Confidence:** 3

**Review:**

In this paper spatio-temporal point convolution are proposed, which can be used for sequences of sparse and unordered data.  For this purpose, a spatial convolution and a temporal convolution are applied separately, which are then combined in a further step.The presented method was evaluated with two different data sets, based on Starcraft II and on data from weather stations.

The method is conclusive and the effectiveness is shown by interesting evaluations. Nevertheless, in my eyes there are still some open questions:
* Why not simply use 4D convolution like for example in the paper Meteornet: Deep learning on dynamic 3d point cloud sequences? In general, the motivation of the method and its components is relatively brief.  For example, an ablation study, where the importance of the time component is shown, would be advantageous.
* The data sets consist of sequences with corresponding point data. What about scan data, without any correspondence? It would be interesting to see a test with semantic segmentation (with lidar data) or action estimation (from depth images) to see more general applications of the TPC layer.
* It is not clear to me why graph convolution networks are not taken into account, in my eyes graph networks would be a good choice for the given data. Recent publications show the abilities of GCNs in similar application areas (e.g. Learning to Simulate Complex Physics with Graph Networks).

The paper itself is written relatively clear and understandable, a few little things I noticed:
* Section 4.3: ... to transform each latent query representation into a final prediction~~**s**~~ ...
* 5.2 Paragraph 3: I think the reference to Figure 6 was confused with the reference to Figure 3.
* Figure 3 and Figure 4: I would leave the scale of the Y-axis of the distribution visualization the same, otherwise it could be misleading.

In summary, I find the method interesting, but in my opinion, it still needs some improvements and more elaboration regarding evaluations. Therefore I tend to a reject.

---

### Official Review · AnonReviewer1 · 2020-10-28
**A simple and effective model for point cloud modelling with irregularity in both space and time**

**Rating:** 4
**Confidence:** 3

**Review:**

This paper studies the problem of modeling spatial-temporal point clouds which are sampled at irregular space and time points. It proposes the Temporal PointConv model which is an extension of the PointConv model (Wu et al., 2019). In particular, PointConv computes a convolution by aggregating the features of nearby points of a point p as the new feature of p.  Temporal PointConv extends this by aggregating the features of points near p in both space and time in a two-step process: first weighting the aggregation by the space distance and then weighting the aggregation by the temporal distance.

Experiments on two datasets show that Temporal PointConv outperforms the baseline models in prediction accuracy.

Pros:
1. The paper studies an interesting topic. Spatial-temporal point cloud modelling has many applications including weather forecasting as shown in the paper.

2. The experimental results are good. The proposed model achieves substantial improvement in terms of the prediction accuracy. The experiments done with a gaming setting (Starcraft II) is interesting.

3. The paper is well written and easy to follow.

Cons:
1. The paper proposed a simple and effective model. However, a concern is on its technical contribution. As discussed above, the proposed Temporal PointConv model is a direct extension of the PointConv model. The technical contribution is too little to justify a publication in a top-tier conference.

Additional comments:
The Abstract claims that the proposed model achieved improved efficiency compared to state-of-the-art baselines, but there is no experimental result reported on model efficiency.

Grammar: "do not have recorded properties we can use to determine property loss." => "do not have recorded properties which we can use to determine property loss."

Typo: "tables 1 and 2" => "Tables 1 and 2"

---

### Official Review · AnonReviewer2 · 2020-10-28
**The approach seems to be problematic and also relies on strong assumptions**

**Rating:** 5
**Confidence:** 4

**Review:**

Summary:

This paper proposes a new spatial-temporal point cloud processing technique, which extends the prior work of PointConv for spatial point processing to the temporal domain. Through experiments on two datasets, this paper shows improved performance over a few baselines.

Paper Strengths:
1. The direction of spatial-temporal point cloud processing is indeed underexplored so I appreciate the efforts put into this direction in the paper.
2. The ability to answer arbitrary prediction queries could be useful in different application domains

Paper Weaknesses:
1. The technical contribution is limited. Essentially, the proposed method is just to apply the existing PointConv operator in both spatial and temporal dimensions. Also, regarding the literature of spatial-temporal point cloud processing, it would be nice if the paper can add some discussion/comparison with PointRNN [A1], which I think is the closest work to this paper though to my knowledge it is not published yet (has been on arXiv for more than a year).
2. The proposed method seems to be problematic in modeling temporal dynamics. Specifically, the PointConv operator in Eq. 1 is to aggregate features within a neighborhood (weighted summation of the features), which does not consider the order of the points in the neighborhood, i.e., permutation-invariant. In the spatial point processing case, this is fine in some applications where we do not consider the direction but only distances of neighborhood points. However, in the temporal domain (i.e., Eq. 3), I feel using PointConv does not make much sense as the order of time should matter in general. Essentially, Eq.3 tells us that, if the centroid point is at frame t, then its corresponding neighbor point at frame t-H has the same distance measure as its corresponding neighbor point at frame t+H, which means that the direction of time is not considered. I feel that the formulation needs to inform the network of the time flow, i.e., the order of time. Also, in Eq 2, the neighborhood also considers some points within a small time window but discards the order of the time again. As a result, I do not think the spatial-temporal point processing proposed in this paper is properly modeling temporal dynamics. Even simple GNN/PointNet + RNN [Li et al 2018b, A2] network structure considers the order of time in the RNN.
3. A big problem I feel about the proposed method is that it relies on a strong assumption about the point correspondences in time, which largely limits the practical impact of this paper. In this paper, the proposed method is shown to reason for high-level entities (e.g., represent an object or a Starcraft unit as a single point), which is fine as the correspondences can be obtained by object tracking in practice. However, how about the scenario where each point does not represent a high-level object but represent low-level observations, e.g., 3D point clouds obtained by LiDAR or stereo reconstruction? In such cases, it is extremely difficult to obtain the point correspondences across time as there could be no correspondences at all in the real-world. Then the proposed method might have problems applied to such data which is widely used in many applications such as point cloud-based object detection, SLAM, point cloud-based classification/segmentation/prediction. In contrast, Minkowski Networks [Choy 2019] that this paper is compared to and also the PointRNN [A1] and SPF [A2] can be applied to these more challenging scenarios.
4. I am a bit concerned about the dataset used in the experiments as the data in the Starcraft II dataset seems to be generated randomly, which is hard to be reproduced and compared by the follow-up work. Would it be possible that evaluating the proposed method on the test set of a few public benchmarks (e.g., traffic prediction) that others can easily compare with even without re-implemented the proposed method?
5. The re-implementation of the Minkowski Networks is somewhat concerning. The paper claims that it is hard to find a suitable kernel size and vocalization resolution, and will lead to a prohibitively high number of network parameters if increasing the kernel size. This leads me to wonder how large scale the data is, e.g., how many points and how many frames the method needs to process. If the scale of the data is not too big, there should not be a problem for the Minkowski Networks I guess. In the original paper of the Minkowski Networks, they evaluated their method on a sequence of large-scale LiDAR point cloud data (usually with 50k number of points per frame), which seems fine and obtains strong performance. It would be nice if a more detailed explanation of why the Minkowski Networks cannot be properly tuned.

Justification:

My decision is made mainly because I feel the proposed method seems to have some flaws and limitations. Also, I am not fully convinced by the experimental results due to the concern about data and re-implementation, and the technical contribution is limited too. However, I would be happy to change my mind if there are any misunderstandings.

References

[A1] H. Fan and Y. Yang. PointRNN: Point Recurrent Neural Network for Moving Point Cloud Processing. arXiv 2019

[A2] Weng et al. Inverting the Forecasting Pipeline with SPF2: Sequential Pointcloud Forecasting for Sequential Pose Forecasting. CoRL 2020

Post-rebuttal Review

As there is no response submitted by the authors, I would like to stick to my original rating to reject this paper.

---

### Decision · Program_Chairs · 2021-01-07
**Final Decision**

**Decision:**

Reject

**Comment:**

The paper proposes a new spatial-temporal point cloud convolution. However, many reviewers suggest the paper can be improved with better baselines and motivations.